# Digital literacy: The catalyst for credit access and income growth in rural households

**Ying Zhang[1,2,3], Haozhao Zhen[1]\*, Yi Wang[4]**

1 School of Economics,Beijing Technology and Business University, Beijing, China, 2 School of Economics and Management, Hainan Normal University, Haikou, China, 3 Zhongshang Guoneng Group Co., Ltd., Beijing, China, 4 School of Labor Economics, Capital University of Economics and Business, Beijing, China

\* 1121942124@qq.com

## Abstract

To clarify the role of digital literacy improvement in farm household income growth, this paper constructs a digital literacy assessment framework covering seven dimensions and uses data from a 2023 rural survey in China to empirically examine the impact of digital literacy on farm household income. The findings indicate that digital literacy significantly enhances income growth among farm households. Mechanism analysis reveals that digital literacy improves access to credit for farm households, thereby facilitating income growth. The effect of digital literacy on household income varies by group, with the highest marginal contribution to agricultural income among low-income households and to non-agricultural income among middle to high-income households. Digital literacy positively affects income across different genders and educational levels. The findings of this study reveal that digital literacy promotes income growth among different groups of rural households in China through various pathways.

## 1. Introduction

2024 marks the 10th anniversary of the strategic goal of building a strong cyber country proposed by the Chinese government. The Chinese government issued the Key Points for Enhancing the Digital Literacy and Skills of the Entire Populationin 2024 [1], which delineates the annual objectives: by the end of 2024, China is poised to ascend to a new level in the development of digital literacy and skills for the entire population, with a more comprehensive system for cultivating digital literacy and skills, comprehensive advancement in the construction of a digital barrier-free environment, and a further narrowing of the digital divide among different groups. The work plan specifies that its core tasks cover 6 areas, focusing on cultivating digital talents, bridging the digital divide, supporting the digital economy, expanding smart life, while strengthening guarantees and cyberspace construction.In 2023, the Chinese

**Data availability statement:** This survey was initiated by the Rural Revitalization Research Institute of Beijing Technology and Business University, and jointly conducted in collaboration with the Department of Agriculture and Rural Affairs of the Ministry of Finance, the Chinese Academy of Fiscal Sciences, and other universities. The Agricultural and Rural Affairs Division of the Provincial Department of Finance was responsible for the distribution and collection of survey materials. As one of the participating parties, we have signed a confidentiality agreement, under which we are not permitted to disclose the survey data from the research report. Any violation of this agreement would subject us to substantial fines and legal consequences.If you need it, please contact the person in charge for communication. After obtaining permission, you can get the data through the link below. In charge: Wei Ziyi e-mail address: weizy0928@163.com Data on the Digital Economy and Common Prosperity in Rural Areas Link: https://pan.baidu.com/s/1eY7K9bKKo5TVstdjunB1cQ.

**Funding:** 1. Beijing Education Science Planning Project (Grant No.: CHAA22060): Research on the Education Model of Improving Digital Literacy of Capital Residents under the Strategy of Common Prosperity. 2. Hainan Provincial Department of Education Higher Education Teaching Reform Research Project (Grant No.: Hnjg2025ZC-46) : Research on the New International Trade Talent Training Model of Hainan Free Trade Port in the Era of Digital Economy. 3. Hainan Provincial Philosophy and Social Science Planning Project (Grant No.: HNSK(YB)24-21): Research on the Mechanism, Effect and Realization Path of the High-Quality Construction of Hainan Free Trade Port Driving Regional Talent Revitalization. The funders had no role in study design, data collection and analysis, decision to publish, or preparation of the manuscript.

**Competing interests:** The authors have declared that no competing interests exist.

government issued the Overall Layout Plan for the Construction of Digital China, proposing that by 2025, digital infrastructure shall be efficiently interconnected, the scale and quality of data resources will be rapidly enhanced, the value of data elements will be effectively released, and the quality and benefits of the digital economy will be significantly improved. The digital countryside represents a strategic orientation for the revitalization of rural China. The Outline for the Development Strategy of Digital Villages issued by Chinese government entities sets strategic objectives: to enhance farmers' digital literacy. This document specifically points out that by 2025, significant progress in digital rural construction is to be achieved; by 2035, substantial advancements are expected, including a marked reduction in the urban-rural digital divide and a notable increase in farmers' digital literacy.These policy documents demonstrate that the Chinese government is implementing measures to enhance nationwide digital literacy, promoting digital economy growth and social digital transformation, with a focus on rural digital advancement [2,3].

As China enters the digital economy era, new-generation information technologies are being increasingly integrated into the modernization of agriculture and rural areas, positioning digital literacy as a pivotal driver of rural household income growth [4]. Households with basic digital proficiency can leverage precision agricultural technologies [5,6]—such as smartphone-based soil testing applications and drone operation systems [7]—to optimize input allocation. Those with advanced digital capabilities further engage in emerging sectors, including rural tourism marketing, paid agricultural knowledge services, and rural e-commerce management [8], thereby diversifying income streams.In the context of the digital economy, the Chinese government has actively advanced the development of digital financial infrastructure [9]. This initiative is particularly relevant as credit constraints persistently impede the optimization of agricultural production inputs, hinder the extension and value addition of agricultural value chains [10], and reduce rural households' risk-bearing capacity [11]. Digital literacy is emerging as a critical mediator enabling rural households to access digital credit mechanisms and alleviate these constraints [12]. Consequently, digital technology penetration facilitates the integration and optimization of resource allocation in rural regions, fostering economic development and income growth. However, the the low level of digital literacy and limited adoption of digital applications among Chinese farmers [13] currently hinder the full realization of the potential benefits of digital technology.

As indicated in The Survey and Analysis Report on Digital Literacy in Rural China released by the Chinese Academy of Social Sciences (CASS) in 2021, the overall digital literacy level of Chinese residents remains relatively low. Specifically, rural residents attained a digital literacy score of 35.1, which is 21.2 points lower than that of urban residents. In terms of digital income-generating capacity, rural residents scored 28.8, representing a 27.7-point gap compared with urban residents. Additionally, their score in terms of digital security awareness stood at 28.6, 43.2 points lower than that of their urban counterparts. As smartphones have been widely popularized in rural areas, rural residents' ability to use such devices is rapidly approaching that of urban residents. These findings indicate that the Chinese government is implementing

measures to enhance nationwide digital literacy, driving the digital transformation of the economy and the digital development of rural areas. While digital devices are widely available in rural areas [14], the value of digital literacy in enhancing rural incomes has yet to be fully realized.

Using survey data from rural China in 2020 or 2021, scholars have found that improved digital literacy enables rural households to better adopt advanced technologies, access market information, expand sales channels, increase employment opportunities [15],broaden social networks, and optimize asset allocation [16], thereby generating a significant ncome-boosting effect. However, data from other countries have shown potential positive correlations [17,18], non-linear correlations [19], or no significant correlations [20] between the popularization and application of digital technologies and individual income.

The impact of rural households' borrowing on their income varies by household income levels and credit types [21,22]. Formal rural credit is subject to the "elite capture" phenomenon, resulting in limited income growth benefits for poor rural households [23],while households with better economic conditions derive greater benefits from credit [24]. Informal micro-credit can iboost income and alleviate poverty, whereas formal rural finance may exert adverse effects [25].

In summary, the changing economic environment and policy support faced by rural households over time, combined with changes in their income levels, lead to uncertain impacts of digital technology and household borrowing on rural households' income. It is therefore necessary to update research findings using the latest data, considering the current income characteristics of Chinese rural households and the latest economic environment they are confronting.

The marginal contribution of this paper are as follows: First, based on UNESCO's "Global Digital Literacy Framework," and in conjunction with the economic development reality of rural China, this paper examines the influence of digital literacy on rural household income across seven dimensions: device and software operations, information and data literacy, communication and collaboration, digital content creation, safety, problem-solving, and career-related competences. Second, while existing literature has theoretically analyzed the pathways of "internet use affecting credit access" and "credit access influencing rural household production," there are few studies exploring the pathway of "digital literacy affecting household income through credit access." By utilizing the latest 2023 survey data from rural China, this paper clarifies the intrinsic mechanism by which digital literacy influences household income via credit access, offering implications for income growth in the digital economy era. Third, while verifying that the enhancement of digital literacy contributes to rural household income growth, this paper also analyzes the heterogeneity of the income-increasing effect of digital literacy—specifically, this effect varies with income levels, educational attainment, and gender.

## 2. Theoretical analysis and research hypotheses

This section investigates the mechanisms through which digital literacy affects rural household income by testing three hypotheses: (1) whether digital literacy increases agricultural and non-agricultural income; (2) whether it boosts income by enhancing access to credit; and (3) whether its impact varies across households categorized by income level, educational attainment, and gender.

### 2.1 The impact of digital literacy on household income

Drawing on Schultz's human capital theory, higher labor quality is correlated with greater labor efficiency, suggesting a positive relationship between human capital investment and income returns. In the digital era, workers' digital literacy has emerged as a crucial form of new human capital [26], which, when enhanced among rural households, can increase their production efficiency [27], thereby increasing income levels. Digital literacy encompasses dimensions including: device and software operations, information and data literacy, communication and collaboration, digital content creation, safety, problem-solving, and career-related competences. Specifically, stronger digital device operation skills enable households to overcome barriers to digital technology access, using digital tools more effectively and fully. Higher digital resource

acquisition literacy allows for better utilization of market information and production factors, reducing job search and business operation costs. Enhanced digital communication and content creation skills enable households to proactively leverage digital technologies for profit, such as engaging in information exchange, product promotion, social capital expansion, and human capital enhancement through online platforms.Greater digital safety literacy fosters risk awareness and seizes on opportunities in new digital technologies and platforms. Stronger problem-solving skills with digital tools improve work efficiency.

In summary, the improvement of digital literacy aids households in better applying advanced agricultural technologies, timely access to market information, and effectively utilizing e-commerce platforms, thereby increasing agricultural income through enhanced production efficiency, reduced operation costs, and expanded product market reach. Additionally, it helps households improve digital skills, expand social networks, enhance risk awareness, and enhance job search capabilities and career competitiveness, which in turn boosts non-agricultural employment and entrepreneurial opportunities.China has made significant strides in rural digital development, leveraging internet technologies to introduce new economic models in rural areas, such as platform economies and flow economies. These innovations have contributed to income growth for rural residents [28,29].

Based on the above analysis, the following hypothesis is proposed

Hypothesis 1: Higher digital literacy significantly increases rural household agricultural income, non-agricultural income, and total income.

## 2.2  Digital literacy influences household income by enhancing loan capacity

Credit funds, as a source of variable inputs, can optimize the initial endowment investment in agricultural production, thereby enhancing agricultural efficiency and reducing income disparities among farmers [30]. In emerging developing countries with a prominent dual economic structure, the contradiction in optimizing the allocation of credit resources between urban and rural areas and across different industries may persist for a certain period.

Traditional formal financial institutions struggle to penetrate rural areas and, to avoid adverse selection and moral hazard, typically impose stringent qualifications on borrowers, leading to the exclusion of farmers. The enhancement of digital literacy allows farmers to utilize digital financial tools [31,32], overcoming geographical constraints and accessing collateral-free, rapid-disbursing credit services [33]. In terms of informal lending, increased digital literacy breaks through the geographical limitations of farmers' social networks, expanding their social circles and enhancing the likelihood of private lending [34]. In formal lending, with improved digital literacy, farmers leave more digital financial usage traces, enabling banks to assess individual creditworthiness based on digital financial information and to observe informal lending behaviors to mitigate information asymmetry in the lending process. Consequently, digital literacy can enhance farmers' credit access, alleviate financial constraints, and alter initial endowments. Access to more credit funds can improve agricultural infrastructure investment, reduce barriers to technology adoption due to insufficient funds, and increase investment in agricultural and non-agricultural operations, production, and sales, thereby promoting income growth for farmers [35,36]. Therefore, the following hypothesis is proposed:

**Hypothesis 2:** Digital literacy promotes household farmer income increase through enhanced credit access.

## 2.3  Heterogeneity of household characteristics

A notable question arises: does the advantage of digital literacy trigger a positive feedback mechanism, leading to a "winner-takes-all" scenario, or what is commonly referred to as the "Matthew effect"? Theoretically, individual endowment disparities may result in unequal application of digital technology, which could, in turn, engender new forms of inequality within the digital economy [37]. Regarding individual endowments, classical growth theory posits that physical and human

 

capital are core determinants of growth. Consequently, the impact of digital literacy on the income growth of farmers is contingent, to some extent, on their own endowments of physical and human capital.

Compared with low-income farmers, high-income farmers possess greater physical capital, which can be invested in reproductive activities and the expansion of production scale. This enables high-income farmers to derive greater economic benefits from the application of digital technology.

The role of human capital in this context remains ambiguous. On one hand, human capital influences the capabilities and cognitive abilities of digital technology application; individuals with higher levels of education may better master the use of digital technologies [38]. On the other hand, individuals with varying educational backgrounds may prefer different digital application scenarios. The network effects and economies of scale in the digital economy have created more profitable models for those with lower educational attainment. For instance, content creators in areas such as short videos and live streaming havedeveloped diverse monetization channels by attracting a large fan base and traffic, including advertising and product endorsements. The Long Tail Effect can aptly explain this phenomenon, describing how a large number of low-unit-value products or services, which individually may not be bestsellers, can collectively form a significant market share, with the aggregate value of these 'long tail' items potentially rivaling or exceeding that of the 'head' (bestsellers). Farmers' participation in markets that align with their cognitive profiles can yield higher returns. The low barriers to entry and high inclusivity of the digital economy have, to some extent, mitigated the traditional impact of educational background on income.

In rural China, traditional norms persist, characterized by the inheritance of physical capital by males and the undervaluation of female human capital accumulation. Such structural gender disparities may lead to heterogeneity in the impact of digital literacy on farmers' income. Under traditional gender division of labor, male farmers tend to engage in large-scale agricultural production—including grain cultivation, cash crop farming, and cross-regional economic activities—whereas female farmers are primarily concentrated in small-scale household agriculture and local service sectors [39]. However, internet usage can mitigate gender discrimination by influencing traditional perceptions and enhancing women's skills and experience [40]. The digital economy boosts women's autonomy and mitigates the "marriage" and "motherhood" penalties [41]. Digital literacy enhances rural women's labor force participation, accumulating their human and social capital to narrow the income gap with rural men.

Hypothesis 3: The marginal impact of digital literacy on household income varies according to different levels of farmers' income, education, and gender.

## 3. Data and methods

### 3.1 Data sources and variable selection

**3.1.1 Source of data.** The data were collected from January to July 2023, initiated by the Rural Revitalization Research Institute of Beijing Technology and Business University in collaboration with 14 institutions, including the Department of Agriculture and Rural Affairs of the Ministry of Finance, the Institute of Fiscal Science of the Ministry of Finance, Anhui University of Finance and Economics, Anhui Normal University, Qingdao University, Jinan University, Jiangxi Normal University, and Northwest Normal University.

Survey Target: The survey was conducted among village party secretaries or village heads, and ordinary villagers in rural administrative villages.

Survey Content: The focus was on digital rural development and shared prosperity in rural areas.

Survey Methodology: The Department of Agriculture and Rural Affairs of the Ministry of Finance selected 30 provinces, with 30 administrative villages randomly chosen from each province. The provincial departments of finance were responsible for distributing and collecting the questionnaires. Universities and research institutions combined the

survey with college students' social practice credits and provided economic and academic incentives to encourage student participation. Surveyors were then selected based on the required survey regions, with clear specifications on the target respondents and the number of surveys to be conducted.

Quality Assurance of the Survey: The questionnaires distributed by the Ministry of Finance were primarily completed through administrative means, with the content mainly derived from statistical data reported by village committees. To ensure data quality, surveyors were required to record audio, take photos with respondents, and provide location of survey sites. The project team conducted random checks on 50% of the questionnaires. Only after these procedures were the questionnaires considered valid.

A total of 1,400 questionnaires were distributed, with 1,283 effectively returned, yielding an effective response rate of 91.64%. The screening criteria included: (1) eliminating extreme outliers; (2) discarding samples with missing basic information; and (3) excluding samples that failed the quality review.

**3.1.2 Dependent variable.** The dependent variable is the total income of farmer's household, encompassing operational, wage, property, and transfer incomes. For empirical analysis, this variable is logarithmically transformed to enhance precision.

**3.1.3 Core explanatory variable.** The core explanatory variable is the digital literacy. The document "A Global Framework of Reference on Digital Literacy Skills for Indicator" by United Nations Educational,Scientific and Cultural Organization(UNESCO) points out that the digital literacy framework includes seven areas of literacy:Devices and software operations,Information and data literacy,Communication and collaboration,Digital content creation,Safety,Problem solving,Career-related competences.

Drawing on the theoretical framework of the "Global Digital Literacy Framework," this paper constructs a secondary indicator system comprising 12 indicators, tailored to the economic characteristics of rural China, as detailed in Table 1. The empirical analysis adopts an equal-weighting scheme in the baseline regression to aggregate the digital literacy levels

**Table 1. Digital Literacy Skills for Indicator.**

| Competency areas | Definition | Mean | SD |
|---|---|---|---|
| Devices and software operations | Possessing a smartphone = 1, otherwise = 0. | 0.560 | 0.247 |
| | Home broadband installation: Installed = 1, Not installed = 0. | 0.944 | 0.053 |
| | Household broadband satisfaction is measured on a scale from 1 to 5, with 1 indicating very dissatisfied and 5 indicating very satisfied. | 3.352 | 1.097 |
| Information and data literacy | Internet shopping participation yes = 1, no = 0. | 0.659 | 0.225 |
| | Digital payment capability: Capable = 1, Incapable = 0. | 0.932 | 0.063 |
| Communication and collaboration | Internet communication and collaboration capability: Capable = 1, Incapable = 0. | 0.912 | 0.080 |
| Digital content creation | Adoption of online marketing tactics: Adopted = 1, Not adopted = 0. | 0.130 | 0.113 |
| | Engagement in live-streaming sales: Engaged = 1, Not engaged = 0. | 0.027 | 0.026 |
| Safety | The Cybersecurity Awareness Scale ranges from 1 to 5, with a score of 1 indicating a lack of familiarity with online anti-fraud knowledge, and a score of 5 representing a comprehensive understanding of online anti-fraud knowledge. | 3.779 | 0.704 |
| Problem solving | Resolution of production issues via internet: Capable = 1, Incapable = 0. | 0.255 | 0.190 |
| Career-related competences | Engagement in online agricultural product sales: Yes = 1, No = 0. | 0.185 | 0.151 |
| | Extent of internet technology application in agriculture and livestock: None = 0, Low = 1, High = 2. | 0.399 | 0.403 |

of farm households. Subsequently, an entropy-weighted approach is used to assign weights and determine the digital literacy levels, thereby assessing the robustness of the equal-weighting results.

**3.1.4 Other control variables.** Control variables include Householder's Characteristic Variables, Household's Characteristic Variables, and Regional Characteristic Variables. The definitions, measurement methods, and descriptive statistics of each variable are provided in Table 2.

Specifically, the household head characteristics include the gender, age, marital status, and education level of the household head; the family characteristics of rural households include the household size, labor force size, operate business, land area, household savings. The regional characteristics include political accessibility, transport accessibility, and village per capita income. In the empirical model, the Village Per Capita Income is logged.

## 3.2 Model setting

To estimate the impact of farmers' digital literacy on their income, this study employs the Ordinary Least Squares (OLS) method as the benchmark regression. Considering the issue of heteroscedasticity in cross-sectional data that may lead to biased parameter estimates, robust standard errors are incorporated in the regression analysis. The model is configured as follows:

$$lnincome_i = \alpha_0 + \alpha_1 DL_i + \alpha_2 X_i + \varepsilon_i \tag{1}$$

where $i$ represents various farm households; $income_i$ denotes the total income of farm household $i$. Here, the log form of total income is used in the estimation; $DL$ denotes the digital literacy of farm household $i$, which is the explanatory variable of interest in this paper. Based on this regression model, the parameter $\alpha_1$ can be utilized to identify the impact of the digital literacy level of rural households on their income. $X_i$ represents the set of control variables, encompassing personal, household, and regional attributes; $\alpha_0$ is the constant term; $\varepsilon_i$ is the error term.

**Table 2. Definition of key variables and descriptive statistics.**

| Classification | Variable Name | Definition | Mean | SD |
|---|---|---|---|---|
| Dependent variables | Total net income | Net annual household income | 114419 | 155765 |
| Core explanatory variables | Digital literacy | Based on the Digital Literacy Skills for Indicator, calculate the weighted score. | 1.011 | 0.207 |
| Control variables | Gender | Householder's gender: Male=1, Female=0 | 0.697 | 0.460 |
| Householder's Characteristic Variables | Age | Age of the Householder | 43.84 | 10.99 |
| | Marital status | Married=1; Other=0 | 0.881 | 0.324 |
| | Educational level | No primary school=1, primary school=2, junior high school=3, high school or technical secondary school=4, vocational school=5, junior college=6, undergraduate =7, postgraduate=8. | 4.766 | 1.584 |
| Household's Characteristic Variables | Household size | Number of people living together in the household | 4.039 | 1.671 |
| | Labor Force Size | Number of Laborers in Households | 2.423 | 1.142 |
| | Operate business | Yes=1,No=0 | 0.355 | 0.479 |
| | Land area | Households' land area | 12.95 | 92.85 |
| | Household Savings | Yes=1,No=0 | 0.663 | 0.473 |
| Regional Characteristic Variables | Political Accessibility | Distance from Residence to County Government (kilometers) | 23.03 | 21.15 |
| | Transport Accessibility | Distance from Residence to the Nearest Railway Station (kilometers) | 43.76 | 45.08 |
| | Village Per Capita Income | Village per capita annual disposable income (in yuan) | 18675.215 | 15509.783 |

To enhance causal inference, this paper employs an instrumental variable (IV) approach to address potential reverse causality and Propensity Score Matching (PSM) to mitigate omitted variable bias. Recognizing the potential heterogeneous effects of digital literacy across income groups, we further apply quantile regression for the heterogeneity analysis.

## 4. Estimation results and analysis

### 4.1 Benchmark regression results

To examine the impact of digital literacy on household farmer income, this paper employs ordinary least squares (OLS) regression based on Equation (1). Robust standard errors are used to correct for potential heteroscedasticity. Regression results are presented in Table 3.

**Table 3. The effect of digital literacy on farm household income.**

| Variables | Income | |
|---|---|---|
| | OLS(1) | OLS (2) |
| Digital literacy | 0.825*** | 0.395*** |
| | (0.104) | (0.0997) |
| Gender | | 0.0642 |
| | | (0.0420) |
| Age | | −0.00163 |
| | | (0.00238) |
| Marital status | | 0.208*** |
| | | (0.0782) |
| Educational level | | 0.0779*** |
| | | (0.0138) |
| Household size | | 0.00920 |
| | | (0.0183) |
| Labor Force Size | | 0.0994*** |
| | | (0.0248) |
| Operate business | | 0.175*** |
| | | (0.0424) |
| Land area | | 0.0124** |
| | | (0.00619) |
| Household Savings | | 0.194*** |
| | | (0.0381) |
| Political Accessibility | | 0.00351*** |
| | | (0.00111) |
| Transport Accessibility | | −0.00153*** |
| | | (0.000493) |
| Village Per Capita Income | | 0.200*** |
| | | (0.0350) |
| Constant | 10.49*** | 7.979*** |
| | (0.104) | (0.380) |
| Observations | 1242 | 1242 |
| Adj.R$^2$ | 0.050 | 0.205 |

Robust standard errors in parentheses.

*** p<0.01, ** p<0.05, * p<0.1.

Columns (1)–(2) in Table 3 examine the association between digital literacy and farmers' household income via a regression framework that incrementally incorporates core explanatory variables and control variables. The adjusted R-squared increases from 0.050 to 0.205, indicating that the addition of control variables improves the model's goodness of fit. Moreover, the coefficients of digital literacy are all significantly positive, indicating that the improvement of digital literacy can significantly increase the overall income of residents, and hypothesis 1 is true. It can be seen from column (2) that the coefficient of digital literacy is 0.395, which is significantly positive at the 1% significance level, indicating that each unit of increase in digital literacy is associated with an approximate 39.5% increase in the farmer's household income. This indicates that in China's all-round development of the digital economy, digital literacy has emerged as a crucial human capital component in enhancing farmers' incomes.

## 4.2. Endogeneity test and robustness analysis

**4.2.1. Endogeneity test.** The baseline OLS estimation may be subject to two types of endogeneity biases that could lead to biased results. First, high-income farmers are more likely to purchase smart devices and participate in paid digital training, which may result in reverse causality bias. Second, there may be omitted variable bias.

This study employs the Instrumental Variable (IV) approach to address reverse causality bias. Specifically, the strategy constructs instrumental variables based on the dual exogenous structure of geographical endowments and historical policies. [42–44]. The first instrumental variable (IV1) selected in this study is the distance from farmers' residences to the administrative center. The location of township-level administrative centers is determined by county-level master plans and has remained stable for a long time since the "township merger policy" in the 1990s, the number of townships decreased from 48,250 in 1992–35,509 in 2005, with an average annual reduction of approximately 2.2% [45]. The farther the distance to the township administrative center, the lower the convenience for farmers to access offline services, and the higher their dependence on digital infrastructure and online digital resources—this indirectly affects digital literacy. The second instrumental variable (IV2) is whether a village has 5G coverage. In 2019, national-level planning launched 5G planning and construction, which did not take farmers' income as the criterion for 5G coverage [46], making it an exogenous variable derived from historical policies. Villages with 5G coverage are more conducive to farmers participating in online digital activities to improve their digital literacy.

We attempt to correct bias using IV. The results of the Two-Stage Least Squares (2SLS) estimation are presented in Table 4. The first-stage regression results show: The overall F-statistic = 14.95 (P < 0.001), which is greater than the empirical critical value of 10, indicating that the two IVs jointly have a significant explanatory power for digital literacy, satisfying

**Table 4. Results of Endogeneity Test.**

| Variables | First-Stage Regression:Digital literacy (1) | Second-Stage Regression:Income (2) |
|---|---|---|
| Digital literacy | | 1.520*** |
| | | (0.543) |
| IV1 | 0.005 | |
| | (0.008) | |
| IV2 | 0.08*** | |
| | (0.011) | |
| Control variables | Yes | Yes |
| Observations | 1242 | 1242 |
| R² | 0.157 | – |

Robust standard errors in parentheses.

*** p < 0.01, ** p < 0.05, * p < 0.1.

the strong correlation assumption. The coefficient of digital literacy is 1.520, which is significant at the 1% level. Compared with the baseline OLS, the coefficient of digital literacy in 2SLS is significantly larger.

We further provide evidence of robustness: As shown in Table 5, the propensity score matching estimates are close to those of the benchmark model, which further validates the conclusion that digital literacy has a positive impact on farmers' income.

The aforementioned results indicate that the coefficient signs of digital literacy on household income, as obtained through the instrumental variable(IV) method and propensity score matching(PSM), are consistent with those of the benchmark model. This suggests that, after addressing endogeneity concerns, digital literacy continues to have a significant positive effect on household income.

### 4.2.2. Robustness test.

1. Replacing the dependent variable. This paper employs household total income as the dependent variable in the baseline regression. Considering that total household income may be influenced by family size, this study replaces the dependent variable with household per capita income and re-conducts the empirical test. As shown in the column(1) of Table 6, digital literacy continues to have a significant positive impact on household per capita income.

2. Weight adjustment. In the baseline regression, an equal-weighting method is used to calculate the digital literacy level of households. The equal-weighting approach is a widely adopted method in the literature for constructing composite indices. This paper further employs the entropy method to recalculate the weights of the digital literacy indicators and conducts another empirical test, with the results presented in the column(2) of Table 6. The regression findings indicate that digital literacy still exerts a significant positive impact on household income.

These tests demonstrate the robustness of the baseline regression results, confirming that digital literacy has a pronounced positive effect on household income.

Table 5. Average Treatment Effect on the Treated (ATT) Estimate.

| Treatment Effect | Treated | Controls | Difference | S.E. | T-stat |
|---|---|---|---|---|---|
| Pre-matching | 11.416 | 11.161 | 0.254*** | 0.043 | 5.79 |
| nearest-neighbor matching | 11.414 | 11.307 | 0.107 ** | 0.047 | 2.26 |
| radius matching | 11.414 | 11.305 | 0.108** | 0.047 | 2.29 |
| kernel matching | 11.414 | 11.304 | 0.110** | 0.047 | 2.33 |

Table 6. Robustness test estimation results.

| Variables | Income | | |
|---|---|---|---|
| | replaces the dependent variable with household per capita income(1) | Entropy weight method(2) | Replace digital literacy measurement(3) |
| Digital literacy | 0.334*** | 0.779*** | 0.128* |
| | (0.104) | (0.160) | (0.0775) |
| Control variables | Yes | Yes | Yes |
| Observations | 1242 | 1242 | 1242 |
| Adj.R$^2$ | 0.291 | 0.212 | 0.196 |

Robust standard errors in parentheses.

*** p<0.01, ** p<0.05, * p<0.1.

3. Substituting the Core Explanatory Variable. In the preceding analysis, a multidimensional composite index was employed to measure digital literacy. To avoid potential biases arising from the method of index synthesis, this study re-examines the empirical relationship using a single indicator—"Internet usage"—as a proxy for digital literacy. The results are presented in column (3) of Table 6. The regression analysis indicates that Internet usage continues to exert a significantly positive impact on household income among rural households.

## 5. Mechanism test

To test the mechanism by which digital literacy increases household farmer income by enhancing credit access capacity, this paper introduces the Loan Capacity variable. First, digital literacy is regressed on Loan Capacity, as shown in Column (1) of Table 7 showing a significant positive impact of digital literacy on Loan Capacity. Subsequently, both digital literacy and Loan Capacity are regressed on farmer's income, as indicated in Column (2) of Table 7 indicating that Loan Capacity acts as a mediating factor, suggesting that digital literacy increases farmer's income by improving Loan Capacity. These results are based on Loan Capacity measured by the "presence of household loans." Additionally, credit access capacity is also measured by the "number of credit access channels," with results consistent with those in Table 7.

Furthermore, this study validates the mediating role of loan capacity through Sobel test and the Bootstrap method. The results of the Sobel test are presented in Table 8: Column 2 indicates the regression of digital literacy on household

**Table 7. Mechanism Test——Alleviating farmers' loan constraints.**

| Variables | Loan capacity(1) | Income(2) |
|---|---|---|
| Digital literacy | 0.179*** | 0.366*** |
| | (0.0672) | (0.0988) |
| Loan capacity | | 0.159*** |
| | | (0.0410) |
| Control variables | Yes | Yes |
| Constant | 0.170 | 7.952*** |
| | (0.230) | (0.382) |
| Observations | 1242 | 1242 |
| $R^2$ | 0.125 | 0.223 |

Robust standard errors in parentheses.

*** p<0.01, ** p<0.05, * p<0.1.

**Table 8. Mechanism Test: Sobel Test Results.**

| Variables | household income(1) | loan capacity(2) | household income(3) |
|---|---|---|---|
| Digital literacy | 0.394*** | 0.179*** | 0.366*** |
| | (0.098) | (0.068) | (0.098) |
| Loan capacity | | | 0.159 *** |
| | | | (0.041) |
| Control variables | Yes | Yes | Yes |
| Observations | 1,242 | 1,242 | 1,242 |
| Adj.$R^2$ | 0.205 | 0.115 | 0.214 |

Robust standard errors in parentheses.

*** p<0.01, ** p<0.05, * p<0.1.

income, Column 3 represents the regression of digital literacy on loan capacity, and Column 4 denotes the joint regression outcomes of digital literacy and loan capacity on household income. The z-value is significant at the 1% significance level.

Utilizing the Bootstrap method with 1000 iterations and a 95% confidence interval, the results in Table 9 indicate that the confidence interval (LLCI = 0.001, ULCI = 0.055) does not include zero, signifying a significant indirect mediating effect with a magnitude of 0.028.

The findings suggest that digital literacy enhances household farmer income by improving credit access capacity, thus validating Hypothesis 2. The cultivation of digital literacy enables farmers to overcome credit constraints imposed by information and technology barriers, expanding credit access channels and increasing credit availability. By leveraging the digital financial data of rural households, credit providers can effectively enhance the efficiency of credit approval, achieve intelligent risk control, and reduce customer acquisition costs. The essence of credit supply and access lies in creditworthiness. On digital platforms, the behavioral data of rural households can be closely integrated with financial services, thereby facilitating the circulation of credit through the information flow of these households.

Specifically, farmers can access material and social capital through internet platforms, which enables them to seek employment or start businesses online. They can generate income by selling their own products, promoting goods on behalf of others, or live-streaming rural life. The data generated from these digital activities serve as the foundation for digital credit assessment on internet finance platforms, thereby further optimizing the credit ratings of rural households. Moreover, internet platforms provide efficient channels for the productive use of credit funds, enhancing the income-generating capacity of rural households. Against the backdrop of China's digital financial development, leading e-commerce platforms have established specialized sections to support agriculture, thereby expanding online sales channels for agricultural products. Meanwhile, social media platforms actively participate in agricultural support initiatives through live-streaming sales and short video promotions. These innovative measures significantly enhance the efficiency and profitability of credit fund utilization among rural households, effectively reducing the risk of default on credit funds and contributing to the optimization of rural households' credit ratings. This, in turn, promotes a virtuous cycle in the supply and access of credit for rural households and provides robust support for the sustainable development of financial services for rural households.

## 6. Heterogeneity analysis

### 6.1 Income level differences

Differences in economic capacity are a key determinant of digital inequality. The development of digital infrastructure has made it possible for low-income groups to access modern digital technologies. However, in terms of the economic output of digital usage, high-income groups are more likely to engage in productive activities with digital technologies (such as work and study), while low-income groups tend to use digital technologies for leisure activities (such as socializing and entertainment), leading to the emergence of digital inequality. Drawing on this analysis, this paper uses household income as a proxy variable to capture variations in the economic capacity of rural households.

Specifically, considering the potential heterogeneity in the impact of digital literacy on household income across different income groups, this paper employs quantile regression analysis, selecting five representative quantiles: 10th, 25th,

**Table 9. Mechanism Test: Bootstrap Test Results.**

| Indirect effect of Loan capacity on Income | | | | | |
|---|---|---|---|---|---|
| Effect | SE | Z | P | LLCI | ULCI |
| 0.028 | 0.013 | 2.06 | 0.040 | 0.001 | 0.055 |
| Direct effect of Loan capacity on Income | | | | | |
| Effect | SE | Z | P | LLCI | ULCI |
| 0.366 | 0.101 | 3.60 | 0.000 | 0.167 | 0.565 |

50th, 75th, and 90th. Table 10 reveals significant differences in the coefficients of digital literacy across quantiles, indicating that the effect of digital literacy on household income exhibits group heterogeneity. Although the impact of digital literacy on total household income is significant and positive across all quantiles, a closer inspection reveals that the coefficient at the 90th quantile is notably higher than at other quantiles. This suggests that while improvements in digital literacy significantly promote income growth for middle- to low-income households, that of the highest-income households more benefits from digital literacy

## 6.2 Income source differences

Table 10 also presents the differences in the impacts of digital literacy on farmers' agricultural income and non-agricultural income. Specifically, digital literacy exerts a significant positive impact on agricultural income, with its marginal contribution to the agricultural income of low-income farmer groups the highest: the coefficient at the 10th quantile is significantly higher than those at other quantiles. This indicates that enhancing digital literacy is beneficial for increasing the agricultural income of low-income households, which could help narrow the income gap within rural households. For non-agricultural income, digital literacy's marginal contribution is greatest for middle- to high-income groups, with coefficients at the 50th and 75th quantiles significantly higher than at other quantiles. These findings suggest that digital literacy can significantly increase agricultural income for low-income groups and non-agricultural income for middle- to high-income groups. This provides insights for both households and policymakers on digital literacy cultivation: focusing on agricultural-related digital skills for low-income groups and non-agricultural industry-related digital skills for middle- to high-income groups, as this targeted skill development offers the greatest marginal contribution to household income enhancement.

## 6.3 Educational level differences

Several studies have demonstrated that individuals with higher levels of education not only possess stronger digital usage skills but also exhibit higher participation rates in digital economic sectors such as e-commerce [47,48]. However, the low barriers to entry and high inclusivity of the digital economy may afford the less educated a "catch-up advantage," enabling them to enter digital industries that align with their cognitive profiles. Particularly in China, e-commerce platforms targeting the "lower-tier market" have generated numerous employment opportunities for individuals with lower human capital [49]. This paper categorizes households into low human capital groups (below high school level) and high human capital groups (high school level and above) based on the educational attainment of the household head to examine the heterogeneous characteristics of education.

The regression results in Table 11 indicate that digital literacy promotes income growth among both low-human-capital and high-human-capital rural households, with a more pronounced effect on the latter.

**Table 10. Heterogeneity test of Income level and Income source.**

| Variables | Income | | | | |
|---|---|---|---|---|---|
| | 10th percentile(1) | 25th percentile(2) | 50th percentile(3) | 75th percentile(4) | 90th percentile(5) |
| Total net income | 0.381*** | 0.305*** | 0.369*** | 0.312** | 0.494*** |
| | (0.140) | (0.105) | (0.0917) | (0.125) | (0.145) |
| Agricultural income | 1.394*** | 1.271*** | 0.656** | 0.571** | 0.502 |
| | (0.334) | (0.322) | (0.255) | (0.260) | (0.399) |
| non-agricultural income | 0.130 | 0.104 | 0.300** | 0.259** | 0.214 |
| | (0.205) | (0.146) | (0.123) | (0.108) | (0.141) |
| Control variables | Yes | Yes | Yes | Yes | Yes |

**Table 11. Heterogeneity test of educational level.**

| Variables | Income | |
|---|---|---|
| | Low human capital(1) | High human capital(2) |
| Digital literacy | 0.344*** | 0.376** |
| | (0.117) | (0.177) |
| Control variables | Yes | Yes |
| Constant | 8.043*** | 6.566*** |
| | (0.506) | (0.683) |
| Observations | 717 | 525 |
| Adj.R$^2$ | 0.237 | 0.171 |

Robust standard errors in parentheses.

*** p<0.01, ** p<0.05, * p<0.1.

This intriguing phenomenon is unfolding in rural China, where individuals of varying educational backgrounds can conveniently engage in live-streaming sales and attract a specific fan base. Influencers with lower educational attainment, often referred to as "grassroots celebrities," can achieve sales volumes comparable to those of "elites" or "stars." In China, the rapid development of the digital economy has created diversified new pathways to income generation for a wide range of groups.

## 6.4 Gender differences

Ordinary Least Squares (OLS) regressions are conducted separately for male and female samples, with results presented in Table 12. Digital literacy exerts a significantly positive impact on the income of both men and women. However, its income-enhancing effect is more pronounced among rural men than among rural women. Despite the fact that digital literacy provides rural women with greater access to employment opportunities, traditional gender role divisions and family responsibilities constrain their participation in high-income sectors. Compared to men, women's entrepreneurship and employment are more concentrated in low-income fields, such as handicraft production and family-based side businesses, which have relatively limited income growth potential.

## 6.5 Differences in digital literacy dimensions

To delve into which aspects of digital literacy have the greatest impact on income, this paper regresses the seven dimensions of digital literacy on household farmer income. Table 13 reveals the impact of each domain of digital literacy. The

**Table 12. Heterogeneity test of gender.**

| Variables | Income | |
|---|---|---|
| | Male(1) | Female(2) |
| Digital literacy | 0.405*** | 0.374* |
| | (0.119) | (0.190) |
| Control variables | Yes | Yes |
| Constant | 8.239*** | 7.469*** |
| | (0.476) | (0.638) |
| Observations | 866 | 376 |
| Adj.R$^2$ | 0.216 | 0.204 |

Robust standard errors in parentheses.

*** p<0.01, ** p<0.05, * p<0.1.

**Table 13. Heterogeneity test of digital literacy dimension.**

| Variables | Income |
|---|---|
| Devices and software operations | 0.092** |
|  | (0.046) |
| Information and data literacy | 0.214*** |
|  | (0.072) |
| Communication and collaboration | 0.023 |
|  | (0.072) |
| Digital content creation | −0.153 |
|  | (0.118) |
| Safety | −0.017 |
|  | (0.024) |
| Problem solving | −0.105** |
|  | (0.048) |
| Career-related competences | 0.273*** |
|  | (0.054) |
| Control variables | Yes |
| Observations | 1242 |
| Adj.$R^2$ | 0.222 |

coefficients for Devices and Software Operations, Information and Data Literacy, and Career-related Competences are significantly positive. This indicates that the primary domains of digital literacy driving income growth for farmers are basic operations of digital devices, information search, and the application of internet technologies related to agriculture. Farmers exhibit significant deficiencies in digital collaboration, digital security, and integrating digital content to solve problems.

## 7. Conclusions and policy implications

With the growth of the digital economy, rural residents can boost their income by improving their digital literacy. Utilizing data from a 2023 survey of Chinese rural households, this paper adopts quantile regression, instrumental variable methods, and propensity score matching (PSM) to examine the impact of digital literacy on household income. The findings are as follows: (1) Digital literacy significantly increases overall household income levels, a conclusion that is robust. (2) Digital literacy enhances farmers' income by improving their capacity to access credit. (3) Digital literacy significantly boosts agricultural income for low-income groups and non-agricultural income for middle- to high-income groups, with the greatest marginal impact on the highest-income households. (4) Digital literacy promotes income growth across individuals with varying levels of educational attainment, yet its impact is weaker among those with lower levels of education. Additionally, digital literacy enhances income growth for both genders, but its positive effect on male income is greater than that on female income.(5) The current dimensions of digital literacy that primarily drive income growth for farmers are basic digital device operations, information search, and the application of internet technologies related to agriculture.

Based on the research, the following policies are proposed: Governments should focus on targeted digital literacy development for rural residents, as this directly supports income growth. For groups where digital literacy has weaker income effects—such as those with lower education levels and rural women—simplified training resources should be developed, including illustrated manuals with QR codes linking to practical videos, and on-site guidance services. These efforts should prioritize building core skills identified as income drivers: basic operation of digital devices, information search, and application of agriculture-related internet technologies. Additionally, to strengthen the channel through which digital literacy boosts income, governments can guide financial institutions to better recognize the value of farmers'

 

improved digital literacy when assessing credit access—helping farmers leverage their enhanced digital skills to secure credit, which in turn supports income growth.

This study has achieved certain results in terms of theoretical analysis and empirical research, but there is still room for improvement. Using only 2023 cross-sectional data may only reflect the short-term impact of both specific digital policies and the post-pandemic economic recovery on rural digital usage. Since the onset of the COVID-19 pandemic, e-commerce has emerged as a primary driver for epidemic control, the resumption of business operations, and the revitalization of the national agricultural and rural economy [50]. Since 2021, the Chinese government has implemented the "Digital Village Development" policy [3], that was reaffirmed in the "No. 1 Central Document" of 2023 and the "15th Five-Year Plan" recommendations for 2025. This policy exhibits sustained consistency and is set to continue its promotion in the future. Furthermore, the pandemic's impact on residents' online consumption habits and digital technology adoption exhibits long-term characteristics [51]. Consequently, while cross-sectional data from 2023 are indeed subject to the dual influences of ongoing policy and post-pandemic economic recovery, these effects are of a long-term nature. This analysis can be validated in future research using time-series data.

Although our survey respondents are predominantly ordinary villagers, a subset of the data was collected from village cadres. These cadres may have performance incentives that lead to an overstatement of digital inputs and outcomes when reporting on items like digital infrastructure and income. This bias could artificially inflate the estimated effect of digital literacy on farmers' income. Future research could attempt to correct for this bias by utilizing more objective metrics, such as backend data from telecommunication operators.

## Author contributions

**Data curation:** Haozhao Zhen.

**Methodology:** Yi Wang.

**Supervision:** Yi Wang.

**Writing – original draft:** Ying Zhang.

**Writing – review & editing:** Haozhao Zhen.

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
