## [Decision Letter · Decision Letter 0]

21 Aug 2025

Dear Dr. Zhen,

Thank you for submitting your manuscript to PLOS ONE. After careful consideration, we feel that it has merit but does not fully meet PLOS ONE’s publication criteria as it currently stands. Therefore, we invite you to submit a revised version of the manuscript that addresses the points raised during the review process.

We look forward to receiving your revised manuscript.

Kind regards,

Dafeng Xu

Academic Editor

PLOS ONE

Journal Requirements:

4. We note you have included a table to which you do not refer in the text of your manuscript. Please ensure that you refer to Table 2 in your text; if accepted, production will need this reference to link the reader to the Table.

Additional Editor Comments:

I have now received two reports, one from a Chinese scholar who is very familiar with the context and methodologies, and one from another country who does not comment on the context but has some insights on the paper's methodologies. I agree with both reviewers that the paper needs at least major revision (in fact, one reviewer rejects the paper). In particular, the authors should think carefully about the IV strategy used in the paper. One of the reviewers mention my two previous papers using the IV; the authors should feel free to read them, but the essence of designing a good IV in this paper's context is to find sufficient exogenous inside an IV, even if such an IV is not perfect. Right now, readers can make easy arguments against the current IV.

Reviewers' comments:

Reviewer's Responses to Questions

**Comments to the Author**

1. Is the manuscript technically sound, and do the data support the conclusions?

Reviewer #1: Partly

Reviewer #2: No

2. Has the statistical analysis been performed appropriately and rigorously?

Reviewer #1: Yes

Reviewer #2: No

3. Have the authors made all data underlying the findings in their manuscript fully available?

Reviewer #1: No

Reviewer #2: No

4. Is the manuscript presented in an intelligible fashion and written in standard English?

Reviewer #1: Yes

Reviewer #2: No

Reviewer #1: Dear Editor

The author constructs a seven - dimension digital literacy assessment framework based on the 2023 China Rural Survey data and empirically examines its impact on farm households' income. With 1,283 valid samples collected by 14 organizations across 30 provinces, the study holds practical significance. I think the paper is acceptable after some modifications.

1.In first the paragraph of the introduction, the focus on rural digital development appears abruptly at the end.

2.Hypothesis 3 refers to differences due to gender levels, but the last paragraph of Section 2.3 mainly argues that the growth of the digital economy has significantly increased overall female employment and income, which is irrelevant to the hypothesis.

3.The equal - weight assignment assumes all indicators contribute equally to digital literacy. But this assumption lacks a solid theoretical basis and isn't verified by actual data. It may underestimate the weight of some key areas and make the measurement results fail to reflect farmers' real digital literacy. Maybe you can consider combined weighting methods like game - theory or grey relational analysis - TOPSIS.

4.The paper positions digital literacy as "a more equitable form of human capital", but how is its fairness reflected? The quantile regression shows digital literacy mostly affects the 90th percentile income farmers. Yet, the paper directly claims "digital literacy promotes fairness", which contradicts the result that "high - income groups benefit more".

5.The author conducts surveys on the Party branch secretaries or village committee directors of rural administrative villages. Reliance on reports from these officials may lead to administrative statistical bias or cognitive errors on household - level data. It's suggested that the author add a discussion on the limitations of data collection methods in the paper.

6.The policy recommendations seem vague and lack feasibility.

7.The reference list doesn't directly cite UNESCO's authoritative reports (such as "A Global Framework of Reference on Digital Literacy Skills for Indicator").

8.Research based on data from a single year is inevitably at risk of being affected by extreme years. The economic policy environment and technological penetration characteristics of 2023 may make the research results time - specific. Although the author has enhanced the reliability of cross - sectional data through rigorous empirical methods, it's necessary to explicitly state the year's limitations in the paper or add time - dimension analysis in follow - up research to strengthen the conclusion's universality.

9.Many viewpoints lack literature support. It is recommended to supplement the literature.

Reviewer #2: This paper studies an interesting question but has some fundamental methodological flaws. I need to note that I am not familiar with China's specific context, so the authors should rely on the editor and other reviewers in terms of improving this part. But at least I can say something about the empirical strategy. My comments:

(1) I don't think OLS is the right model to study this research question. While the authors correctly refer to it as a "benchmark model," the main focus of the paper should be on the IV regression. I won't even introduce the equation for OLS in Section 3.

(2) I don't believe that the aggregate IV is the right model either. This variable, in this paper's empirical setting, unnecessarily creates a network problem because a person's digital literacy (and socioeconomic status) will be undoubtedly affected by the neighbors' digital literacy and socioeconomic status, unless the authors are ready to claim that individuals in this paper's empirical setting are all very isolated (but how so??). Charles Manski has a famous "reflection problem" paper entailing this, so I don't think that this IV is an appropriate one here.

The least the authors can do is to design a historically driven aggregate IV, which is still problematic but is at least more plausible than using a current measure to quantify a current level of literacy (and other outcomes). For example, some historical measures of infrastructure, human capital, or average household characteristics dated back to decades ago from the statistics yearbook. This way, the authors make a more plausible IV -- but keep in mind that even if the authors do so, it would be extremely important to contextually convince readers why such an IV is plausible, and it requires careful explanation for the Chinese context. I know very little about China, so I'd rather stop commenting here.

I believe that the editor himself has a few papers about it: see his "Acculturational homophily" (Econ of Ed Review) and Identifying ethnic occupational segregation (J of Pop Econ). I hope he can further navigate the authors on this matter.

(3) Relatedly, the authors did not mention anything about the OLS bias. Of course the OLS is biased. But how biased? Why is it upward (or downward) biased? What factors drive this bias, and is there an indirect way to evaluate it? The authors did a poor job analyzing the exact way endogeneity emerges in the dataset.

(4) I would just drop PSM. It doesn't help much solve endogeneity problems.

(5) Overall, I identify a lot of writing issues. The authors may want to re-work the writing. I am also not a native speaker, so I know it is hard. But the authors can surely do a little bit more to make the paper more readable.

**Do you want your identity to be public for this peer review?** For information about this choice, including consent withdrawal, please see our Privacy Policy

Reviewer #1: No

Reviewer #2: No

---

## [Author Response · Author response to Decision Letter 1]

31 Oct 2025

Dear Editor:

This article has been revised regarding the financial disclosure statement to align with the funding information. If you need to make changes in the submission system, please modify according to the content in the financial disclosure statement.

Title: Digital Literacy: The Catalyst for Credit Access and Income Growth in Rural Households.

Article Number: PONE-D-25-29153R1.

---

## [Decision Letter · Decision Letter 1]

26 Nov 2025

Dear Dr. Zhen,

Thank you for submitting your manuscript to PLOS ONE. After careful consideration, we feel that it has merit but does not fully meet PLOS ONE’s publication criteria as it currently stands. Therefore, we invite you to submit a revised version of the manuscript that addresses the points raised during the review process.

We look forward to receiving your revised manuscript.

Kind regards,

Dafeng Xu

Academic Editor

PLOS ONE

Journal Requirements:

Additional Editor Comments :

This round, we only receive one review, so I myself also read the paper carefully. Overall the paper looks fine, but both the reviewer and me believe that the paper should improve the literature review throughout the paper, as well as writing issues. I am attaching a .doc file with my comments, and the reviewer also present their feedback. Please revise the paper and provide responses to these comments in the next round.

Reviewers' comments:

Reviewer's Responses to Questions

**Comments to the Author**

Reviewer #1: (No Response)

2. Is the manuscript technically sound, and do the data support the conclusions?

Reviewer #1: Yes

3. Has the statistical analysis been performed appropriately and rigorously?

Reviewer #1: I Don't Know

4. Have the authors made all data underlying the findings in their manuscript fully available?

Reviewer #1: No

5. Is the manuscript presented in an intelligible fashion and written in standard English?

Reviewer #1: Yes

Reviewer #1: Dear Editor,

The author has addressed most of the issues, and the scientific rigor of the article has been improved. However, there are still some issues that need further discussion:

1.Regarding the analysis that only uses 2023 data, since the Editor and the other reviewer have no further comments, I also choose to accept this conclusion. However, the limitations of using 2023 data are not clearly stated. It is advisable to conduct discussions on whether there were special digital policies in 2023 and the short-term impact of post-pandemic economic recovery on rural digital usage, etc.

2.For the revision, it is supplemented that the survey subjects include ordinary villagers (the original statement only mentioned village cadres). However, it does not address the limitations of data bias, does not discuss the possible administrative statistical bias in the reports of village cadres, and nor does it analyze the potential impact of this bias on the research results.

3.Many viewpoints lack literature support, and it is recommended to supplement relevant literature. Only 2 articles have been added since the last manuscript, which is unreasonable for such a lengthy discussion in this paper. Upon careful review, many viewpoints are the author's own opinions. For example, the first sentence should be supported by the reference:

Cyberspace Administration of China, et al. (2024). 2024 work priorities for improving national digital literacy and skills. Retrieved from http://www.gov.cn.

For instance, line 64 can introduce the literature by Liu et al. to elaborate on digital economy and agricultural development: Liu, F., Zheng, D., Zhao, D., & Wen, B. (2026). Spatiotemporal differentiation and evolution of associated networks of China's agricultural innovation ecosystem. Agricultural Systems, 231,104537. https://doi.org/10.1016/j.agsy.2025.104537. Lines 67-68 can draw on Xu’s article for support related to digital finance: Xu, K. (2023). Digital finance, social security expenditures, and rural-urban household income poverty. Evidence based on an area and household level analysis. Finance Research Letters.

**Do you want your identity to be public for this peer review?** For information about this choice, including consent withdrawal, please see our Privacy Policy

Reviewer #1: No

---

## [Author Response · Author response to Decision Letter 2]

3 Jan 2026

Dear academic editors and reviewers,

We would like to extend our sincere gratitude to all of you for your respective valuable comments and constructive suggestions on our manuscript. These insights have been instrumental in enhancing the quality and rigor of our work, and we have carefully addressed each comment from reviewer and academic editor in the revised version.

---

## [Decision Letter · Decision Letter 2]

6 Jan 2026

Digital Literacy: The Catalyst for Credit Access and Income Growth in Rural Households

PONE-D-25-29153R2

Dear Dr. Zhen,

We’re pleased to inform you that your manuscript has been judged scientifically suitable for publication and will be formally accepted for publication once it meets all outstanding technical requirements.

Kind regards,

Dafeng Xu

Academic Editor

PLOS One

Additional Editor Comments (optional):

Reviewers' comments:

Reviewer's Responses to Questions

**Comments to the Author**

Reviewer #1: All comments have been addressed

2. Is the manuscript technically sound, and do the data support the conclusions?

Reviewer #1: Yes

3. Has the statistical analysis been performed appropriately and rigorously?

Reviewer #1: Yes

4. Have the authors made all data underlying the findings in their manuscript fully available?

Reviewer #1: Yes

5. Is the manuscript presented in an intelligible fashion and written in standard English?

Reviewer #1: Yes

Reviewer #1: Dear Authors,

I believe that after this round of revisions, you have effectively addressed the previous concerns. Therefore, I recommend the acceptance of this study. Wish you a happy New Year!

**Do you want your identity to be public for this peer review?** For information about this choice, including consent withdrawal, please see our Privacy Policy

Reviewer #1: No

---

## [Editor Report · Acceptance letter]

PONE-D-25-29153R2

PLOS One

Dear Dr. Zhen,

I'm pleased to inform you that your manuscript has been deemed suitable for publication in PLOS One. Congratulations! Your manuscript is now being handed over to our production team.

Kind regards,

on behalf of

Dr. Dafeng Xu

Academic Editor

PLOS One